# Determinants of timely administration of the birth dose of hepatitis B vaccine in Senegal in 2019: Secondary analysis of the demographic and health survey

Oumar Bassoum [1,2] *, Mouhamadou Faly Ba[3], Ndeye Mareme Sougou [1,2], Adama Sow[1,2], Ibrahima Seck[1,2]

1 Department of preventive medicine and public health, faculty of medicine, pharmacy and odontology, Cheikh Anta Diop University, Dakar, Senegal, 2 Institute of Health and Development, Cheikh Anta Diop University, Dakar, Senegal, 3 Cheikh Anta Diop University, Dakar, Senegal

* oumar.bassoum@ucad.edu.sn

**Data Availability Statement:** The datasets analyzed during the current study are accessible on https://dhsprogram.com/.

## Abstract

In developing countries, hepatitis B is spread primarily by the perinatal or horizontal route. Thus, the Senegalese government recommends administering the birth dose of the hepatitis B vaccine (HepB0) within 24 hours of birth. The objective was to identify the determinants of timely administration of HepB0 in Senegal in 2019. A secondary analysis of the demographic and health survey was carried out. The study population consisted of children aged 12 to 23 months. A cluster survey, stratified in urban and rural areas, drawn at two stages, was carried out. Individual interviews were conducted. Logistic regression was applied to estimate the adjusted odds ratio (aOR) and their 95% confidence interval. In total, 1130 children were included. Among them, 48.1% were born to mothers aged 25 to 34, 46.8% were male and 82.6% were born in health facilities. The average time between birth and HepB0 administration was 13.1±46.6 days. The median is 0 days [IQR: 0–12] with a minimum of 0 days and a maximum of 414 days. Among 747 children vaccinated, only 65.2% were vaccinated within 24 hours of birth. The determinants were maternal age of at least 35 years (aOR = 2.03 [1.29–3.20]), primary education of the mother (aOR = 1.94 [1.13–3. 35]), at least four antenatal care visits (aOR = 1.74 [1.12–2.69]), belonging to the central (ORa = 0.22 [0.11–0.44]) and northern regions (aOR = 0.18 [0.08–0.40]), and delivery in a health facility (aOR = 3.42 [1.90–6.15]). Education and keeping girls in school, local postnatal care in hard-to-reach regions, promotion of antenatal acre and delivery in a health facility should improve the timeliness of HepB0 vaccination.

## 1. Introduction

Hepatitis B virus (HBV) infection is a public health concern, with an estimated 296 million people chronically infected and 820 000 deaths worldwide in 2019 [1]. The severity of the infection is linked to the transition to chronicity which can be complicated by cirrhosis or

**Funding:** The authors received no specific funding for this work.

**Competing interests:** The authors have declared that no competing interest exist.

hepatocellular carcinoma which are mainly responsible for deaths [2]. Another fact that explains the worrying nature of HBV infection is that several studies have shown gaps in terms of knowledge and practices of the population regarding the disease [3, 4] or vaccination at birth [5]; thereby limiting engagement and demand for vaccination services [6].

Hepatitis B endemicity is classified according to the seroprevalence of hepatitis B virus surface antigen (HBsAg). It is classified as low (<2%), low to intermediate (2% to 4.9%), intermediate high (5% to 7.9%), and high (≥ 8%) [7, 8]. In areas of high endemicity, HBV is transmitted mainly at the time of childbirth (perinatal transmission) or during early childhood (horizontal transmission) [7]. However, infection in infants exposes them to a high risk of chronicity. In such a context, WHO recommends that all infants, including those who are low birth weight or premature, receive their first dose of hepatitis B vaccine (HepB0) as soon as possible after birth, preferably within 24 hours [9]. Hepatitis B vaccination is an essential care that should take place between 90 minutes and 6 six hours after birth after weighing the newborn [10].

Timely administration of HepB0, i.e. within 24 hours of birth, should serve as a performance indicator for any expanded program on immunization (EPI) [9]. Indeed, the administration of HepB0 within this time frame is not only an entry point into the EPI but also a quality indicator. Despite this, the vaccination time is relatively neglected. Vaccination targets are largely focused on gross coverage of recommended vaccines before a certain age without regard to whether these are administered at the appropriate age or not [11].

The fight against hepatitis B is at the heart of the 2030 agenda [12]. It is expected to reduce new infections by 90% and mortality by 65% compared to 2015 [13]. To this end, the "Global health sector strategies against, respectively, HIV, viral hepatitis and sexually transmitted infections for the period 2022–2030" aim to increase the percentage of newborns benefiting from HepB0 in time intended and other interventions aimed at preventing mother-to-child transmission of HBV to 70% in 2025 and 90% in 2030 [14].

In Senegal, according to a modeling study, the seroprevalence of HBsAg is 8.1% (95% confidence interval: 7.5%-9%) [15]. Therefore, it is a country of high endemicity [16]. From 1999 to 2004, vaccination against hepatitis B was carried out as part of the national hepatitis control program. In 2005, vaccination is carried out within the framework of the EPI in a combined form called pentavalent which protects against diphtheria, tetanus, whooping cough, *Haemophilus influenzae* type b (Hib) infections and hepatitis B. Pentavalent is administered at 6, 10 and 14 weeks of age. Then, in 2016, HepB0 (monovalent vaccine), was introduced into the EPI [17]. The vaccine comes in liquid form in a ten-dose vial [17]. Depending on the opened vial policy, the vaccine can be used for a maximum of four weeks after opening. Conservation takes place at a temperature between +2°C and +8°C. The dose is administered intramuscularly in the anterolateral part of the left thigh [18].

The 2019–2023 comprehensive multi-year plan (cMYP) aims for HepB0 vaccination coverage greater than 90% by 2023 [19]. The demographic and health survey (DHS) conducted in 2019 revealed that HepB0 coverage was 81.3% [20]. Two studies carried out respectively in Podor and Niakhar estimate vaccination coverage within 24 hours of birth at 42.1% in 2020 and 66.8% in 2017–2018 [21, 22]. Factors negatively associated with HepB0 vaccination are home delivery, delivery in the dry season, and delivery in 2016 [22]. Favorable factors are delivery in a health facility, access to television, weighing immediately and hospitalization of the newborn immediately after birth [21]. These studies have shortcomings. First, the DHS does not explicitly mention whether it is vaccination coverage within 24 hours or not. Second, the geographic scope of the studies done in Podor and Niakhar is limited; making the generalizability of the results questionable. Thus, it appeared necessary to fill these gaps through this

study whose objective is to highlight the determinants of the timely administration of HepB0 in Senegal.

## 2. Materials and methods

### 2.1. Study framework

Senegal, located in West Africa, is between 12˚8 and 16˚41 north latitude and 11˚21 and 17˚32 west longitude. Its surface area is 196,722$^{km2}$. In 2019, the population was estimated at 16,209,125 inhabitants. Children aged 0–4 years accounted for 16.39% [23]. Vaccination is provided to children according to a fixed strategy (population residing within a health facility), advanced strategy (population residing within a radius of between 5 and 15 km) and mobile strategy (population residing within a radius greater than 15 km) [17].

### 2.2. Type of study

This was a secondary analysis of DHS data conducted in 2019 [20]. The DHS is a cross-sectional survey. The data was collected throughout Senegal, between April and December 2019 with a 30-day break, i.e. a period of eight months of collection. DHS are carried out under the DHS Program established by the United States Agency for International Development (USAID) in 1984 [24].

### 2.3. Study population and sampling

The study population consisted of children aged 12 to 23 months residing in Senegal at the time of the survey. Eligible subjects were those who had spent the night preceding the survey in the selected household [25]. Complex formulas are used for DHS sampling. A procedure using SAS software is used for the calculation following an appropriate statistical methodology. This procedure uses the Taylor linearization method for estimates such as averages or proportions, and the Jackknife method for more complex estimates (such as indices). This was a two-stage, stratified, cluster sampling. Each region was separated into urban and rural parts to form the sampling strata. Senegal has 14 regions. Thus, 28 strata were created. Sampling is carried out independently within each stratum. At the first stage, 214 census districts (CD) were drawn with a probability proportional to their size. This was the number of resident households in the CD. The list of CD is established during the General Census of Population and Housing, Agriculture and Livestock (RGPHAE) carried out in 2013. At the second level, in each of the CD selected at the first level, 22 Households were selected by the coordination team using systematic equal probability sampling. Replacements of pre-selected households were not authorized on the spot, even for non-responding households, to avoid selection bias.

In summary, the necessary number of clusters was 214. In each of them, 22 households were selected according to a systematic sampling with equal probability in both urban and rural areas. The number of households is 4,708 including 1,848 in urban areas and 2,860 in rural areas. The sample size was 1130 children aged 12 to 23 months [20].

### 2.4. Collection of data

Data collection was conducted in two ways: a face-to-face interview with the mother or caregiver and consultation of the child's health record. It was carried out by 20 interviewers trained in the technique of administering questionnaires using tablets. The team members exchanged data using Bluetooth while the transfer of data to the servers, by the team leaders, was done via the Internet. At the end of each day, team leaders had to transfer the data to the central server [20].

### 2.5. Variables studied

The dependent variable is timely administration of HepB0. It is defined as the administration of HepB0 within 24 hours of birth. It is binary (HepB0≤24 hours/HepB0>24 hours) [26]. The choice of this operational definition is explained by the fact that in Senegal HepB0 is recommended from birth, preferably within 24 hours following birth [17]. The independent variables relate to the individual and the context. The individual characteristics are: age of the mother, mother's education level, marital status of the mother, birth order, sex of the child. The contextual characteristics are: area of residence, type of residence, household size, sex of household head, wealth quintile, father's education level, antenatal care (ANC), tetanus toxoid (TT) injection, place of birth, mode of delivery, checking of the child's health before leaving the health facility, possession of home-based record (HBR).

### 2.6. Statistical analyzes

All results have been obtained considering weighting and the complex sample design. The data were described in the form of numbers and percentages (qualitative variables) but also in the form of mean±standard deviation (quantitative variables). The chi-square test with Rao and Scott's correction was used to compare the proportions of children who received HepB0 on time according to the levels of the independent variables. Then, logistic regression was applied with the variables for which the p-value is less than 0.25 during the univariate analysis. The top-down step-by-step procedure has been completed. The collinearity hypothesis was tested but was not significant. All the analyzes are carried out with the R software.

### 2.7. Ethical considerations

The survey protocol and questionnaires were sent to the National Ethics Committee for analysis and approval. Verbal consent was obtained before inclusion. Verbal consent is justified by the high illiteracy rate among Senegalese women, especially in rural areas. The person surveyed was informed of the non-obligatory nature of participation and of the possibility of interrupting the survey at any time. Data collection was anonymous and confidential. Access to the database was authorized by "The Demographic and Health Surveys (DHS) Program". The data were accessed for research purposes on 18 May 2021. Authors had no access to information that could identify individual participants during or after data collection.

## 3. Results

### 3.1. Participant characteristics

The children were born to mothers aged under 35, illiterate and married in 76.0% (859), 60.1% (679) and 94.8% (1,072) of cases respectively. The male gender represented 53.2% (602). In addition, 30.9% (349) resided in the western part of the country and 62.1% (702) in rural areas. Births in health establishments reached 82.6% (933) (S1 Table).

### 3.2. Vaccination coverage

Among the 1130 children included, 921 received HepB0, representing crude coverage of 81.5%. However, only 747 children had a vaccination card indicating the date of vaccination. The average time between the day of birth and that of vaccination is 13.1±46.6 days. The median is 0 days [IQR: 0–12] with a minimum of 0 days and a maximum of 414 days. Among them, 65.2% (487/747) were vaccinated on time, i.e. within 24 hours of birth (Fig 1).

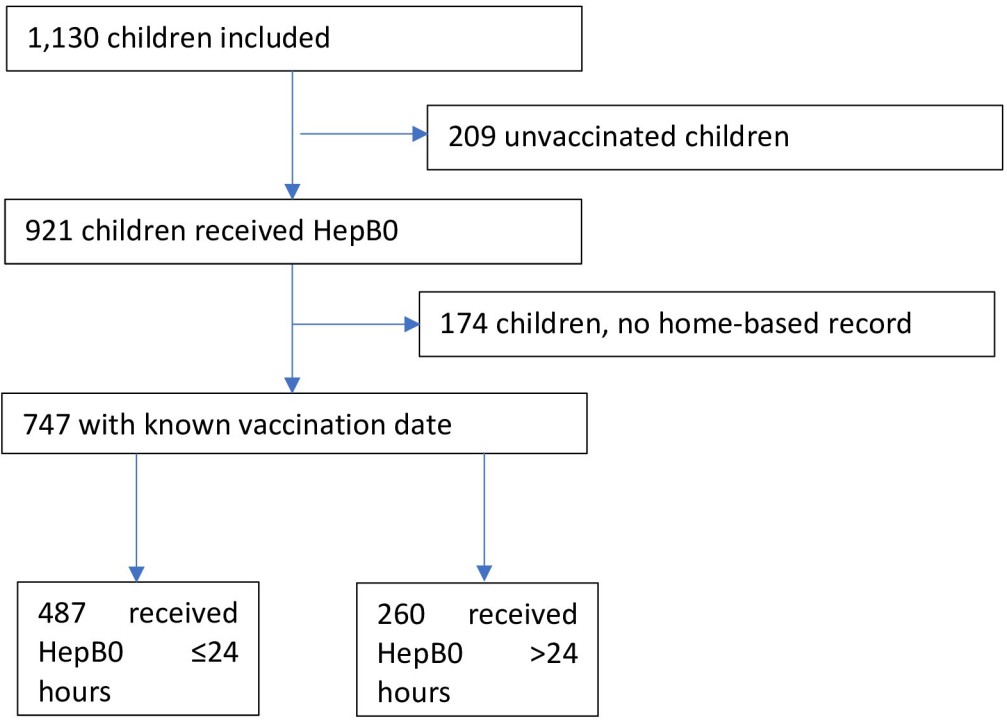

**Fig 1. Flow chart of vaccination status with HepB0.**

### 3.3. Bivariate analysis

The results of the bivariate analysis indicate that the percentages of children vaccinated within 24 hours of birth varied significantly depending on the age of the mother, the mother's education level, the area of residence, the type of residence, wealth quintile, number of ANC, place of birth, exposure to media such as newspapers and television, access to mobile phones and access to the Internet (S2 Table).

### 3.4. Binary logistic regression

The multivariable analysis showed that individual characteristics such as age $\geq$ 35 years and maternal primary education increased the probability of timely vaccination of the child by 2.03 respectively [1.29–3.20] and 1.94 [1.13–3.35]. As for contextual characteristics, regions located in the center and the north reduced the chance of getting vaccinated on time by 78% (0.22; [0.11–0.44]) and 82% (0.18; [0.08–0.40]). On the other hand, this chance increased by 1.74 [1.12–2.69] and 3.42 [1.90–6.15] times when the child was born to a mother who had completed at least four ANC or was born in a health facility (S3 Table).

## 4. Discussion

### 4.1. Vaccination coverage

Vaccination coverage within 24 hours of birth is lower than that recommended in the cMYP which is 90%. It is higher than those that have been highlighted in the world (42%) and in Africa according to administrative data (6%) [27] and surveys (1.3%) [28] and 21.2% for Nigeria [29]. These results highlight the challenges facing the Senegalese health system in implementing timely HepB0 vaccination. As a result, the road ahead to achieve the goal of

eliminating hepatitis B by 2030 is long. This present study identified factors associated with timely administration of HepB0.

## 4.2. Determinants

Maternal age of at least 35 years was significantly associated with vaccination within 24 hours of birth. This may be because older mothers may have experience and better knowledge of the effect and importance of vaccination than younger ones. An identical result was demonstrated in Ethiopia [30]. On the other hand, raising awareness among young mothers through social networks is necessary because it has been shown that the use of these communication tools to constantly disseminate reliable information creates a strong image of vaccination, thus improving coverage vaccination [31]. Maternal age 25–34 years was not significantly associated with vaccination within 24 hours of birth, compared with at least 25 years.

Maternal education level predisposes to timely vaccination. Indeed, children whose mothers had a primary education were almost twice as likely to receive HepB0 within 24 hours compared to their counterparts whose mothers were not educated. In Ethiopia, a similar result was found [32]. In China, the higher level is a favorable factor compared to the primary level. In contrast, in the United States and Israel, the opposite effect was observed because children born to more educated mothers were less likely to receive HepB0 on time [33]. These two situations are contradictory. However, they reveal the significant role of education in access to care. Strategies may vary from one context to another. In developed countries, the schooling rate for women is high. They are exposed to disinformation online; thus leading to vaccine hesitancy. On the other hand, in developing countries, women are less educated and therefore less exposed to social networks. Therefore, promoting vaccination will involve education and keeping girls in school.

This study showed that at least four ANC increases the chance of receiving HepB0 on time by 1.74. This result clearly shows the importance of prenatal education on the prevention of hepatitis B from birth. In Senegal, the 2019-DHS shows that 98% of women received ANC provided by a trained provider [20]. But only 55.5% received at least four. This situation calls for two actions. The first is that women should be made aware of the importance of using ANC services according to national recommendations. Next, healthcare professionals should be trained to address hepatitis B birth vaccination during ANC [26]. According to a qualitative study, Nigerian women attending ANC services suggest that health professionals include education on HepB0 during ANC [34].

Furthermore, the study revealed that birth that takes place in a health facility triples the probability of receiving HepB0 on time. Several studies have shown low coverage of primary vaccination at birth when births take place outside a healthcare facility [33]. Childbirth, particularly that carried out by qualified personnel, constitutes an opportunity not only to administer HepB0 but also to provide advice on the importance of birth registration and compliance with the vaccination schedule [10]. In a context marked by a high rate of home childbirth, the WHO recommends two actions. The first is to make home visits to provide postnatal care, including administration of HepB0. The second action is to bring newborns to the health facility as soon as possible. However, Boisson A et al. showed that geographic distance prevents the timely delivery of newborns to health facilities. Therefore, the second option involves raising awareness among pregnant women during ANC or in the community using community health workers [33]. According to the 2019-DHS, the percentage of births that took place in a health facility is 80.3% [20]. This result, although encouraging, suggests a missed opportunity for vaccination against hepatitis B and could be explained by a lack of communication between vaccination services and maternal health services.

Area of residence is another factor associated with timely administration of HepB0. Children from distant regions, located in the north and center, are disadvantaged compared to those residing in regions located to the west such as Dakar and Thiès. The latter are better off in terms of health infrastructure and human resources in health [23]. In Nigeria, a similar result was found [35]. Overall, this study highlights the existence of inequity in timely vaccination because coverage varies significantly in terms of social and geographic characteristics. This inequity could be avoided through structural measures by improving geographic access to vaccination services [36].

### 4.3. Strengths and limitations of the study

The study is distinguished by three main strengths. First, the sufficiently large sample size and the prior determination of the number of clusters and the number of households to be surveyed improve the precision of the estimates. The second strength is the low risk of selection bias because the subjects were selected, not by the investigators, but by a coordination team. The selection was made according to a random procedure guaranteeing representativeness not only at the national and regional level but also according to the urban and rural area of residence. Finally, the risk of information bias is also low because the vaccination status and date of vaccination are extracted from the vaccination card.

On the other hand, the study has limitations. First, the consultation of the health facility registry was not carried out contrary to the WHO recommendation. This led to the exclusion of several children whose vaccination status was neither recorded nor dated in the HBR. The second limitation is that the data were not collected specifically for this study; making it impossible to identify other determinants of timely vaccination. Finally, the variable "wealth quintile" could not be quantified because the DHS data does not provide the wealth thresholds to classify households into quintiles. This lack of quantification is a limitation for the analysis of the influence of wealth level on child vaccination.

### 5. Conclusion

This study shows that HepB0 vaccination coverage at the appropriate age remains significantly lower than crude vaccination coverage, suggesting that using the latter as the main indicator is likely to overestimate actual protection. Timely vaccination should therefore be considered as an indicator of vaccination performance.

Given the limitations of the study, further additional studies with all three sources (maternal recall, register, card) and contextual variables would be necessary.

## Supporting information

**S1 Table. Characteristics of participants, DHS, 2019, N = 1130.**
(DOCX)

**S2 Table. Bivariate analysis of timely HepB0 vaccination, DHS, 2019, N = 747.**
(DOCX)

**S3 Table. Multivariate analysis of significant determinants of timely HepB0 vaccination, DHS, 2019, N = 747.**
(DOCX)

## Acknowledgments

We thank "The Demographic and Health Surveys (DHS) Program" for authorizing access to the database. We also thank the 'Société africaine de santé publique' for allowing us to present this study at its first congress held from November 15 to 17 in Grand Bassam, Ivory Coast.

## Author Contributions

**Conceptualization:** Oumar Bassoum, Mouhamadou Faly Ba.

**Data curation:** Mouhamadou Faly Ba.

**Formal analysis:** Oumar Bassoum, Mouhamadou Faly Ba.

**Methodology:** Oumar Bassoum.

**Supervision:** Ndeye Mareme Sougou, Ibrahima Seck.

**Writing – original draft:** Oumar Bassoum, Adama Sow.

**Writing – review & editing:** Oumar Bassoum, Mouhamadou Faly Ba, Ndeye Mareme Sougou, Ibrahima Seck.

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
