## [Decision Letter · Decision Letter 0]

24 Jan 2024

PGPH-D-23-02359

Determinants of timely administration of the birth dose of hepatitis B vaccine in Senegal in 2019: Secondary analysis of the demographic and health survey

Dear Dr. Bassoum,

Thank you for submitting your manuscript to PLOS Global Public Health. After careful consideration, we feel that it has merit but does not fully meet PLOS Global Public Health’s publication criteria as it currently stands. Therefore, we invite you to submit a revised version of the manuscript that addresses the points raised during the review process.

We look forward to receiving your revised manuscript.

Kind regards,

Orvalho Augusto, MD, MPH

Academic Editor

Journal Requirements:

Additional Editor Comments (if provided):

This is quite an important topic. Prevention of vertical transmission of hepatitis B needs to be improved in SSA. The authors present a secondary analysis of DHS 2019 Senegal data to assess the timeliness of the Hepatitis B vaccine and find its correlates. However, there are important technical issues in the analysis.

1. Was the complex survey sampling addressed in this analysis? The current statistical section says nothing about this. As the reviewer rightly points out there please provide more details on the statistical procedures used here.

2. Table 2 uses bivariate p-values and no measure of association such as OR or RR or RD. Please put the measure of association and its 95%CI.

Please try to put them into single table 1 and table 2.

3. Table 3: It does not make sense to include the same variable in one categorization in table 2 then in table 3 use another. For example, education primary versus none. Where are are other categories?

4. Please reorganize the discussion section. Move the strengths and limitations of the study to the end of discussion section.

Reviewers' comments:

Reviewer's Responses to Questions

**Comments to the Author**

1. Does this manuscript meet PLOS Global Public Health’s publication criteria? Is the manuscript technically sound, and do the data support the conclusions? The manuscript must describe methodologically and ethically rigorous research with conclusions that are appropriately drawn based on the data presented.

Reviewer #1: Yes

2. Has the statistical analysis been performed appropriately and rigorously?

Reviewer #1: Yes

3. Have the authors made all data underlying the findings in their manuscript fully available (please refer to the Data Availability Statement at the start of the manuscript PDF file)?

Reviewer #1: Yes

4. Is the manuscript presented in an intelligible fashion and written in standard English?

Reviewer #1: Yes

5. Review Comments to the Author

Reviewer #1: Re: Determinants of timely administration of the birth dose of hepatitis B vaccine in

Senegal in 2019: Secondary analysis of the demographic and health survey

Thanks for allowing me to revise this important topic on hepatitis B birth dose, a secondary analysis of DHS 2019 in Senegal

General comments: Though generally well written, there a few comments for authors

Abstract: “In total, 1130 children were included. Among them, 48.1% were born to mothers aged 25 to 34, 46.8% were male and 82.6% were born in health facilities. The average time between birth and HepB0 administration was 13.1±46.6 days. Among 747 children vaccinated, only 65.2” - Unclear, you included 1130, 747 were vaccinated? you are interested in timeliness ?

“13.1±46.6 days.” Present as median with IQR, add range.

Introduction: “A recent study estimates that 296 million people will be chronically infected and 820,000 will die worldwide in 2019 [1]- Kindly recast? we are 2024!!! If it is projected, recast to reflect past event on projection

Methods: Kindly include sample size estimation?

“Model adequacy was analyzed using the Hosmer-Lemeshow test.” No information on this in the results?

Results: Participant characteristics-include n before the percentage.

Vaccination coverage: present the measure of central tendency with median plus IQR, add range as well.

“Among them, 65.2% (487/747) were vaccinated on time, i.e. within 24 hours of birth” why is this not the actual sample size?

Multivariate analysis-change to binary logistic regression

Multivariate is actually multivariable analysis-your outcome is still binary-the only thing is that you entered multiple variables!!!

Table 3: There are quite a number of variables that were significant on 2 by 2 in Table 2, they were not included in Table 3?

Vaccination coverage: It will be nice to compare your data with a large country with similar challenges in the immunization coverage -see Assessment of the Timely Administration of Birth Dose Vaccines in Northern Nigeria and Associated Factors (Ann Glob Health. 2022; 88(1): 60.Published online 2022 Jul 26. (doi: 10.5334/aogh.3743)

6. PLOS authors have the option to publish the peer review history of their article (what does this mean?). If published, this will include your full peer review and any attached files.

**Do you want your identity to be public for this peer review?** For information about this choice, including consent withdrawal, please see our Privacy Policy.

Reviewer #1: **Yes: **Olayinka Ibrahim

---

## [Decision Letter · Decision Letter 1]

9 Apr 2024

PGPH-D-23-02359R1

Determinants of timely administration of the birth dose of hepatitis B vaccine in Senegal in 2019: Secondary analysis of the demographic and health survey

Dear Dr. Bassoum,

Thank you for submitting your manuscript to PLOS Global Public Health. After careful consideration, we feel that it has merit but does not fully meet PLOS Global Public Health’s publication criteria as it currently stands. Therefore, we invite you to submit a revised version of the manuscript that addresses the points raised during the review process.

We look forward to receiving your revised manuscript.

Kind regards,

Orvalho Augusto, MD, MPH

Academic Editor

Journal Requirements:

Additional Editor Comments (if provided):

Dear co-authors,

The questions/comments from the Editor were not responded. I paste here those questions.

=======

This is quite an important topic. Prevention of vertical transmission of hepatitis B needs to be improved in SSA. The authors present a secondary analysis of DHS 2019 Senegal data to assess the timeliness of the Hepatitis B vaccine and find its correlates. However, there are important technical issues in the analysis.

1. Was the complex survey sampling addressed in this analysis? The current statistical section says nothing about this. As the reviewer rightly points out there please provide more details on the statistical procedures used here.

2. Table 2 uses bivariate p-values and no measure of association such as OR or RR or RD. Please put the measure of association and its 95%CI.

Please try to put them into single table 1 and table 2.

3. Table 3: It does not make sense to include the same variable in one categorization in table 2 then in table 3 use another. For example, education primary versus none. Where are are other categories?

4. Please reorganize the discussion section. Move the strengths and limitations of the study to the end of discussion section.

======

Reviewers' comments:

Reviewer's Responses to Questions

**Comments to the Author**

1. If the authors have adequately addressed your comments raised in a previous round of review and you feel that this manuscript is now acceptable for publication, you may indicate that here to bypass the “Comments to the Author” section, enter your conflict of interest statement in the “Confidential to Editor” section, and submit your "Accept" recommendation.

Reviewer #1: All comments have been addressed

2. Does this manuscript meet PLOS Global Public Health’s publication criteria? Is the manuscript technically sound, and do the data support the conclusions? The manuscript must describe methodologically and ethically rigorous research with conclusions that are appropriately drawn based on the data presented.

Reviewer #1: Yes

3. Has the statistical analysis been performed appropriately and rigorously?

Reviewer #1: Yes

4. Have the authors made all data underlying the findings in their manuscript fully available (please refer to the Data Availability Statement at the start of the manuscript PDF file)?

Reviewer #1: Yes

5. Is the manuscript presented in an intelligible fashion and written in standard English?

Reviewer #1: Yes

6. Review Comments to the Author

Reviewer #1: (No Response)

7. PLOS authors have the option to publish the peer review history of their article (what does this mean?). If published, this will include your full peer review and any attached files.

**Do you want your identity to be public for this peer review?** For information about this choice, including consent withdrawal, please see our Privacy Policy.

Reviewer #1: No

---

## [Editor Report · Decision Letter 2]

1 May 2024

PGPH-D-23-02359R2

Determinants of timely administration of the birth dose of hepatitis B vaccine in Senegal in 2019: Secondary analysis of the demographic and health survey

Dear Dr. Bassoum,

Thank you for submitting your manuscript to PLOS Global Public Health. After careful consideration, we feel that it has merit but does not fully meet PLOS Global Public Health’s publication criteria as it currently stands. Therefore, we invite you to submit a revised version of the manuscript that addresses the points raised during the review process.

We look forward to receiving your revised manuscript.

Kind regards,

Orvalho Augusto, MD, MPH

Academic Editor

Journal Requirements:

Additional Editor Comments (if provided):

Dear authors,

I am confused with the tables. It seems that the old tables (without the changes I suggested) are still there. Please, please revise.

---

## [Editor Report · Decision Letter 3]

10 May 2024

PGPH-D-23-02359R3

Determinants of timely administration of the birth dose of hepatitis B vaccine in Senegal in 2019: Secondary analysis of the demographic and health survey

Dear Dr. Bassoum,

Thank you for submitting your manuscript to PLOS Global Public Health. After careful consideration, we feel that it has merit but does not fully meet PLOS Global Public Health’s publication criteria as it currently stands. Therefore, we invite you to submit a revised version of the manuscript that addresses the points raised during the review process.

We look forward to receiving your revised manuscript.

Kind regards,

Tinsae Alemayehu, MD

Guest Editor

Journal Requirements:

Additional Editor Comments (if provided):

Thank you for your submission.

Please address the following items.

- Please remove the tables from the main text as they are bulky and submit them as supplements

- Some categories in table 2 are not clear: What does "checking the child's health before leaving the health facility" represent?

- (Table 2) Please use standard medical terminology for mode of delivery

- (Table 2) Describe abbreviations at the bottom of the table

- (Table 2) How is household lead determined?

- (Table 2) Quantify the wealth quintiles

- (Table 2) Elaborate “area of residence” – are those the names of regions?

- Include missing comparison categories into the multivariate analysis discussion in table 3
---

## [Editor Report · Decision Letter 4]

13 Jun 2024

PGPH-D-23-02359R4

Determinants of timely administration of the birth dose of hepatitis B vaccine in Senegal in 2019: Secondary analysis of the demographic and health survey

Dear Dr. Bassoum,

Thank you for submitting your manuscript to PLOS Global Public Health. After careful consideration, we feel that it has merit but does not fully meet PLOS Global Public Health’s publication criteria as it currently stands. Therefore, we invite you to submit a revised version of the manuscript that addresses the points raised during the review process.

We look forward to receiving your revised manuscript.

Kind regards,

Tinsae Alemayehu, MD

Guest Editor

Journal Requirements:

Additional Editor Comments (if provided):

Please address the following items.

- Please remove wealth quintiles if they cannot be quantified. Or describe this gap as a limitation for analysis by clearly outlining why the variable cannot be quantified.

- Please use formal region names instead of crude axis directions, which are subjective for analysis.

- If you have listed ONLY significant multivariate analysis in table 3, the title should say so
---

## [Editor Report · Decision Letter 5]

25 Jul 2024

Determinants of timely administration of the birth dose of hepatitis B vaccine in Senegal in 2019: Secondary analysis of the demographic and health survey

PGPH-D-23-02359R5

Dear Dr Bassoum,

We are pleased to inform you that your manuscript 'Determinants of timely administration of the birth dose of hepatitis B vaccine in Senegal in 2019: Secondary analysis of the demographic and health survey' has been provisionally accepted for publication in PLOS Global Public Health.

Best regards,

Tinsae Alemayehu, MD

Guest Editor